# 3D Bioprinting Technology and Hydrogels Used in the Process

**DOI:** 10.3390/jfb13040214

**Published:** 2022-11-03

**Authors:** Tainara de P. L. Lima, Caio Augusto d. A. Canelas, Viktor O. C. Concha, Fernando A. M. da Costa, Marcele F. Passos

**Affiliations:** 1Graduate Program in Materials Science and Engineering, Federal University of Pará, Ananindeua 67130-660, Brazil; 2Technological Development Group on Biopolymers and Biomaterials from the Amazon, Institute of Biological Sciences, Federal University of Pará, Belém 66077-830, Brazil; 3School of Chemical Engineering, Federal University of São Paulo, Diadema 09913-030, Brazil; 4Institute of Biological Sciences, Federal University of Pará, Belém 66075-110, Brazil

**Keywords:** 3D bioprinting, hydrogels, tissue engineering

## Abstract

3D bioprinting has gained visibility in regenerative medicine and tissue engineering due to its applicability. Over time, this technology has been optimized and adapted to ensure a better printability of bioinks and biomaterial inks, contributing to developing structures that mimic human anatomy. Therefore, cross-linked polymeric materials, such as hydrogels, have been highly targeted for the elaboration of bioinks, as they guarantee cell proliferation and adhesion. Thus, this short review offers a brief evolution of the 3D bioprinting technology and elucidates the main hydrogels used in the process.

## 1. Introduction

Three-dimensional bioprinting (3DBP) is a multidisciplinary area of interconnection between life sciences and engineering. Through the combination of cells, growth factors, and biomaterials, and the principles of additive manufacturing, 3DBP emerges as a promising technology for use in tissue engineering and regenerative medicine [1,2]. It is, in fact, a therapeutic alternative in the development of materials that mimic the structure, composition, and function of native tissues. Some injured organs or tissues have a limited ability to regenerate, and their physicochemical and biological characteristics are heterogeneous and complex. Thus, 3D bioprinting technology uses different biomaterials to meet biological and mechanical functionalities. Recent advances have highlighted, in particular, hydrogels as a raw material in 3DBP [3,4,5].

Hydrogels are three-dimensional polymeric networks that have a high capacity to absorb fluids without dissolving [6,7,8], being well established as scaffolds in the area of tissue engineering (TE) [9,10]. Due to their hydrophilic nature and design with a porous structure, they tend to support cellular growth, proliferation, and differentiation. They can also be vehicles for biologically active substances or cells [2,11]. However, the successful scientific results with 3DBP hydrogels, of natural or synthetic origin, are usually fragile, with limited mechanical tenacity [12]. Research demonstrates, for example, that hydrogels for use in cartilage tissue engineering, have a fracture energy in J/m^2^ that is ten times lower than the fracture energy of natural cartilage (~1000 J/m^2^). However, with 3D bioprinting of hydrogels, the architecture of these materials, in terms of geometry, interconnection, and pore size, can be adjusted, integrating other mechanisms of mechanical reinforcements [9]. Such properties play a crucial role in intercellular signaling, enhancing the development of macroscopically functional living constructs.

### 3D Bioprintng Evolution

Bioprinting is an emerging and multidisciplinary technology that originated from 3D printing (additive manufacturing) (Figure 1). The first milestone, in 1984, was provided by Charles W. Hull, through the development of three-dimensional printing objects (3D), via stereolithography (SLA). In 1988, the researcher Robert J. Klebe used cytoscribing technology to demonstrate the potential of positioning biological products, using a Hewlett Packard (HP) inkjet printer and a graphic plotter [13,14,15]. Years later (1999), David J. Odde and Michael J. Renn printed living cells using 3D laser bioprinting, thus, demonstrating the feasibility of synthesizing tissues with complex three-dimensional anatomies [16,17]. In the 2000s, Rolf Muelhaupt and his group reported the first three-dimensional plotting of thermosensitive gels in a liquid medium, using the additive manufacturing technique [18]. Later, in 2002, the first extrusion-based bioprinter was reported by Landers et al., being marketed under the name “3D-Bioplotter” [19,20]. In 2003, Boland et al. adapted an HP inkjet printer and were able to successfully print living cells [21]. In 2006, Suwan N. Jayasinghe and his team added an electro-hydrodynamic jet to deposit living cells [22]. In 2009, Narotte et al. synthesized vascular tissue based on free scaffolds [23]. In 2012, Skardal and colleagues performed in situ bioprinting in laboratory mice using cells derived from amniotic fluids to stimulate the healing process. The results indicated that the bioprinting of these cells could be used to treat wounds and burns [24]. Several types of research have been developed to generate new products for society and overcome the challenges of 3D bioprinting. Zhou et al. (2021), for example, used 3D bioprinting technology to add chondrogenic progenitor cells (CPCs) and fibronectin (FN) to a hydrogel composed of alginate/gelatin/hyaluronic acid (Alg/Gel/HA), intending to optimize the cartilage regeneration process [25]. Nulty et al. (2021) developed a new bioprinting method for manipulating pre-vascularized tissues in vitro to analyze vascularization and bone regeneration in vivo [26]. Ramasamy et al. (2021) synthesized an artificial skin using an extrusion-based 3D bioprinter. This research aimed to identify an opportunity to provide full-thickness reconstructed human skin in a reproducible and potentially scalable manner [27]. Noor et al. (2019) synthesized a custom hydrogel for printing autonomous cellular structures, such as whole hearts with blood vessels [28].

## 2. 3D Bioprinting: Concept and Characteristics

Three-dimensional bioprinting has recently been used as an excellent alternative in the field of tissue engineering and regenerative medicine to develop biological substitutes, scaffolds, in vitro drug models, and artificial organs or tissues [29]. The main objective is to customize complex biological structures in a reproducible and fast manner, combining cells, biomaterials (matrices), and growth factors [19,30,31]. The process consists of the successive and automated addition of living and non-living materials, with a structural organization using computer-aided design (CAD) programs. Generally, the 3D bioprinting technique can involve the following steps (Figure 2): Acquisition and processing of medical imaging data; Three-dimensional bio-modeling and/or description of the required tissue/organ geometry; Formulation of inks from biomaterials (without cells) or cells added to synthetic and/or organic materials (bioinks); Three-dimensional bioprinting (calibration and slicing); Maturation; Physical–chemical, and biological, analysis of the bioprinted structure [32,33]. Furthermore, this technology has several advantages, such as geometric freedom and control, precision, automation, repeatability, customization, and the generation of constructs with the potential to capture responses to external stimuli and physiological functions [34].

### Biomaterial Inks and Bioinks

Biomaterial inks are cell-free aqueous formulations that contain biological factors and are usually made of polymers or hydrogels [35]. Subsequently to the printing step, the 3D constructs are seeded, and they seek to mimic the extracellular matrix of target tissues, with a favorable environment for cell adaptation and proliferation [36,37]. The applications are often directed towards the developing of scaffolds and implants or can be used in parallel with the fabrication of bioinks in hybrid approaches, ensuring mechanical support [38,39]. One of the examples of inks from biomaterials is sacrificial materials (such as agarose, Pluronic 127, alginate, and gelatin), which are printed and dissolved without affecting cell survival [40,41]. Compaan et al. (2016) used alginate as a sacrificial material during the jet bioprinting process [42]. Biodegradable thermoplastic polymers, such as polycaprolactone (PCL) and poly(lactic acid) (PLA), and non-biodegradable thermoplastic polymers, such as polypropylene, are also commonly used [43,44]. Bioinks are formulations containing cells, biomaterials, and growth factors, and they can be classified as scaffold-based and scaffold-free bioink, respectively. The technique based on scaffolds is used much more in 3D bioprinting, since living cells are encapsulated in the matrix components of the biomaterials and, later, bioprinted in a pre-elaborated structure [45]. On the other hand, the scaffolds-free bioink technique use multicellular spheroids and cell aggregates to be bioprinted in a 3D model [46,47]. Furthermore, bioinks are defined in four subcategories: Support bioinks, which serve as an extracellular matrix for cell multiplication; Fugitive bioinks, which are temporary materials, and, which, when removed, can form an internal void; Structural, which provide sustainability to printed structures; Functional, which provide biochemical, electrical and mechanical stimuli, optimizing cell behavior [35]. In work by Hu et al. (2020), a bioink composed of chitosan grafted with polyethylene glycol (PEG), α-cyclodextrin (α-CD), and gelatin was synthesized to be applied in tissue and organ remodeling [48]. Zhang et al. (2021) developed a bioink based on silk fibroin and decellularized extracellular matrix (SF-dECM), incorporated with mesenchymal stem cells from bone marrow, for use as scaffolds in cartilage tissue engineering [49]. In addition, in the research by Jian et al. (2021), it was possible to optimize the preparation of bioinks based on gelatin methacrylate (GelMA) and meniscal extracellular matrix (MECM) for scaffold bioprinting [50]. Constructs obtained from decellularized extracellular matrix (dECM) bioinks mimic the physical and mechanical microenvironment of native tissues and organs, combining cells and biochemical signals, such as growth factors and cytokines, and the right proportions of extracellular matrix (ECM) proteins. The great advantage is being able to control the position and placement of cells, in addition to preserving most ECM components [51]. Decellularization is like a reservoir of various molecules, removing cells from native tissues via chemical, physical or mechanical processes. The dECM bioprinting approach is a personalized therapeutic approach for tissues and organs, as the bioink has specific composition and topology [52]. However, the exclusive use of dECM bioinks has the disadvantage of low mechanical stability, and, therefore, they are mixed with other polymers or support materials [53].

Some required characteristics of bioinks and/or biomaterials ink for use in 3D bioprinting are non-toxicity, mechanical resistance [54], printability, in vitro and in vivo cell viability [55], structural stability [56], viscosity [57], rheological properties (rheopexy/thixotropy), and surface tension [58,59]. These variables directly impact cell encapsulation and the format in which the material is printed [60,61]. Moreover, the bioink must present fluidity, enabling three-dimensional printing, have a printing temperature not higher than the physiological temperature, and proper gelation kinetics to form a solid structure.

## 3. Bioprinting Technologies

To print tissues and organs with greater complexity, targeting applications in tissue engineering and regenerative medicine, several 3D bioprinting technologies have been developed and optimized, as can been seen in Table 1 [55,62]. Each technology, however, is limited by the properties of the bioinks and influence the bioprinting quality of the material [63]. High viscosities, for instance, require extrusion bioprinters, as it favors a moderate flow capacity, conserving the structure of the printed materials for a more extended period [64]. Low-viscosity bioinks are suitable for a jet bioprinter as they can be ejected easily through a fine nozzle without too much pressure [65]. In addition to these two types of technologies (extrusion and inkjet), laser-assisted bioprinting is also highlighted [66].

### 3.1. Inkjet Bioprinting

Inkjet bioprinters (drop-on-demand printers) are capable of printing biological materials with optimized speed, accuracy, and resolution (Figure 3) [77]. They can work with single and multi-ink systems, printing materials with precision and geometric complexity [32]. Furthermore, they utilize thermal and acoustic (piezoelectric) forces to deposit liquid droplets of defined size, layer by layer [78]. In the case of thermal force, rapid electrical heating is provided in the bioprinter head, which generates pressure pulses that force the droplets out through the nozzle. This heating can vary between 200 °C to 300 °C without causing damage to the cells [79,80]. The piezoelectric forces generate an acoustic wave causing the pressure necessary to eject the drop from the nozzle [81]. For this class of bioprinter, the aim is to use bioinks with low viscosity and cell density. Moreover, bioink gelation must be carried out in situ to avoid nozzle clogging, which is one of its disadvantages [81,82,83]. Yerneni et al. (2019), for example, synthesized solid-phase exosomes using a piezoelectric jet bioprinter, aiming at localized delivery of exosomes into tissues [84].

### 3.2. Extrusion-Based Bioprinting

Extrusion-based bioprinting is one of the most used bioprinting techniques today, as it prints bioinks with high viscosity [55,85,86]. In this process, the bioinks are extruded as a thread through the nozzle, using two methods: pneumatic (air) and mechanical (piston and screw) (Figure 4) [77]. In the pneumatic method, the air pressure provides the force to eject the bioink with pre-established speed and quantity [87]. However, even though it is a simple procedure, there is a lack of control in bioinks with low viscosity [88]. The mechanical method determines the printing process through the vertical and rotational forces [79,81]. In the piston process, the flow is favored over the bioink during printing. For very viscous materials, however, failures occur in the bioink deposition. On the other hand, in screw-based extrusion, bioink is distributed in a microliter range, which can be interesting for materials with low viscosity [89]. Although extrusion bioprinting is one of the most required for artificial tissues and organs, it has some disadvantages, such as shear stress, which can cause death and/or loss of cell viability, and a low quantity of materials [90,91]. It is recommended to use more robust hydrogels and improvements in the nozzle and syringe, which would contribute to better cell viability after printing [54]. In work by Cleymand et al. (2021), a bioink based on chitosan (CH) and guar gum (GG) was synthesized to be used in extrusion-based bioprinters [92].

### 3.3. Laser Assisted Bioprinting (LAB)

This method does not use a nozzle and contact. The laser passes through a region of three layers: transparent, absorbent, and bioink (Figure 5) [55]. The transparent region supports the absorbent layer, and the biomaterials are in liquid/gel physical condition to spread more easily [93]. Three techniques make up this method: matrix-assisted pulsed laser evaporation direct writing (MAPLE-DW), where a low-power pulsed laser with an ultraviolet wavelength is used; laser-induced transfer (LIFT), which uses a high-power pulsed laser and a thin absorbent layer between the donor slide and the bioink; and film-assisted laser-induced direct transfer absorption (AFA-LIFT), which uses a thick absorbent layer that prevents direct interaction between the laser and the bioink [94]. Furthermore, LAB has a minimal effect on cell viability, and can print hydrogels with varying viscosities [87].

## 4. Hydrogels as Bioinks or Bioprinting Ink

The use of cells or cell aggregates as bioinks, in 3D bioprinting technologies, especially in extrusion-based printing, must deal with the challenge of the low cellular viability of the constructs, due to cells finding it difficult to resist the shear stress caused by the material’s deposition process, layer by layer [1]. Alternatively, hydrogels act as vehicles for encapsulating and delivering cells, maintaining high shape fidelity, and mimicking the native extracellular matrix [2,11]. Furthermore, these materials have good biodegradability and biocompatibility. Designed as porous structures, they present a promising microenvironment for gas exchange and nutrient diffusion for 3D bioprinting of cells [53]. The swelling capacity provided by the three-dimensional network of hydrogels becomes fundamental for cell migration, proliferation, and adhesion, enhancing the development of complex tissues and organs [95,96,97,98]. Thus, several efforts have been devoted to formulating hydrogel-based bioinks for a 3D microenvironment suitable for cell seeding and encapsulation.

The selection of hydrogel (synthetic or natural) as a bioink is intimately related to the bioprinting technique, tissue type and selected cells. Furthermore, its formulation must satisfy rheological and biological criteria. The viscosity, concentration and crosslink density must also be optimized [9], with three main types of crosslinking used in the post-printing procedure: thermal, chemical or physical (UV light, among others) [71]. Other parameters, such as cytotoxicity, printability [99], physical strength [51], in vitro or in vivo degradation capacity and the effects of by-products in the culture medium are also important for the reproduction of good materials [100]. Among the most used hydrogels as bioinks or bioprinting inks are the following: hyaluronic acid, alginate, silk-fibroin, collagen, agarose, and gelatin. Table 2 presents the advantages and disadvantages of each hydrogel used as a bioink.

### 4.1. Hyaluronic Acid

Hyaluronic acid is a linear non-sulfated glycosaminoglycan in most connective tissues and the extracellular matrix [120,121]. This natural hydrogel exhibits excellent biocompatibility, hydrophilicity, and cytocompatibility in cell development [122,123,124,125]. However, its low mechanical property generates the need for crosslinking with other polymers to satisfy the physicochemical properties of 3D bioprinting [62]. Photopolymerization—crosslinking materials in the presence of ultraviolet (UV) rays is also an excellent alternative to increase the mechanical strength of hyaluronic acid [9,126,127,128]. In work by Antich et al. (2020), a new type of bioink was developed through the copolymerization of hyaluronic acid with polylactic acid (PLA) for use in 3D bioprinting of tissues and cartilage [101]. Kiyotake et al. (2019) synthesized a pentanoate-functionalized hyaluronic acid (PHA) bioink, using the photocrosslinking (UV) procedure, to be employed in 3D bioprinting [129].

### 4.2. Collagen

Another widely used natural hydrogel and one of the most attractive is type I collagen. It stands out because it is one of the components in musculoskeletal tissue and makes up the extracellular matrix of other tissues [130,131,132,133]. Furthermore, collagen presents biocompatibility, biodegradability, and cell adhesion, all of which are fundamental properties in the field of 3D bioprinting [134]. However, like hyaluronic acid, collagen has a low mechanical property, which is improved with photopolymerization (UV) [135,136]. Evidence of this is found in the work of Shi et al. 2018. In this research, the authors developed a new type of bioink through the photopolymerization process between collagen and methacrylated gelatin hydrogel (GelMa), which might be an excellent candidate for 3D bioprinting of tissues to repair damage to the epidermis [137].

### 4.3. Gelatin

Gelatin is a polypeptide obtained through collagen denaturation [72]. It has been studied for the development of bioink due to its properties, such as biocompatibility [115], biodegradability, low cost, ease of processing, and cell affinity [118]. Furthermore, gelatin can optimize its mechanical properties when crosslinked with other materials, such as methacrylic anhydride [73,138]. Several kinds of research have focused on printing this functionalized form of gelatin (Gelatin-methacryloyl/GelMA) to obtain better material parameters. An example of this is the work of Jain et al. (2021). In this study, the authors investigated GelMA bioinks loaded with mouse fibroblast cells (L929) to be used in an extrusion-based bioprinter. The results indicated that the material would be a good candidate for the synthesis of constructs with and without the presence of a cell [139]. Silk fibroin is another material with which gelatin can be combined [140]. In the research by Singh et al. (2019), a bioink based on gelatin–silk fibroin was synthesized. The materials demonstrated good print fidelity, which suggested a high potential to act in the repair of cartilage tissues [141].

### 4.4. Alginate

Alginate is a natural polysaccharide composed of β-D-mannuronic acid (M) and α-L-glucuronic acid (G) [142,143]. It is widely used in 3D bioprinting due to its biocompatibility, printability, affordable prices, and versatility [65]. Furthermore, its ease of gelation in the presence of divalent cations (Ca^+2^ and Ba^+2^, for example) optimizes the structural form of the construct and minimizes the effect of shear stress on cells, which has favored its application in inkjet and extrusion bioprinting [115]. The rheological parameter of alginate must be carefully analyzed when applied in bioprinting, since the viscosity of the bioink based on this hydrogel is directly linked to the concentration, molecular weight of the alginate, phenotype, and cell density [114,144]. Pure alginate presents some difficults to promote cell proliferation and weak mechanical properties. However, these disadvantages can be changed when it is mixed with other materials [145]. Wu et al. (2018) developed an alginate/cellulose-based hybrid bioink (CNCs) to be used in the extrusion bioprinting process. The results demonstrated a good shear property of the material, maintenance of the shape of the construct, and minimal cellular damage [146]. Lee et al. (2020) synthesized and supplemented an alginate-based bioink with decellularized methacrylated extracellular matrix (dECM) derived from bone tissues. The researchers concluded that the material could print structures loaded with 3D cells and maintain cell viability [145].

### 4.5. Agarose

Agarose is a natural polysaccharide extracted from seaweed, being composed of D-β-galactose (D-Gal) and 3,6-anhydro-α-L-galactose (L-AHG) [147,148,149]. It has a limitation for cell proliferation, but this can be overcome when mixed with other hydrogels. Furthermore, it can serve as a template material for 3D cell aggregate culture [62,117]. Fan et al. (2016) developed a hybrid agarose/matrigel system with favorable rheological properties for 3D bioprinting. The results showed that agarose contributed to the support of the printed structure, while the matrigel provided a favorable environment for cell growth [150]. López-marcial et al. (2018) analyzed the rheological properties, such as storage modulus and shear stress, of agarose and alginate-based hydrogels, which were compared with Pluronic F-127. The alginate/agarose-based materials showed excellent cell viability, indicating their applicability as a bioink [116].

### 4.6. Silk Fibroin

Another interesting material for the elaboration of bioinks is silk fibroin (SF) [151]. Produced from Bombyx mori (B. mori), also known as the silkworm, this material has attracted attention in the 3D bioprinting segment, due to its excellent properties, such as biodegradability [152], biocompatibility [153], processing in different formats (hydrogels, films, membranes, etc.) and good mechanical properties [154]. Furthermore, when it comes to the manufacture of bioinks, another parameter that must be evaluated is the viability of cell growth [155]. In this regard, SF also stands out, as it has different viability for different types of cell lines, good encapsulation of cells and bioactive compounds, in addition to being approved by the Food and Drug Administration (FDA). However, polymers from natural origins may present rheological disadvantages and the need for mixing with other polymeric materials [65,156]. In the work by Wei et al. (2021), a bioink based on silk fibroin, gelatin, hyaluronic acid and tricalcium phosphate, interspersed with human platelet-rich plasma (PRP), was developed with potential use in bone tissue engineering [157]. In the research by Kim et al. (2021), a bioink, with optimized cytocompatibility, composed of alginate and silk fibroin (Alg/SF), was synthesized to be used in tissue engineering [158].

## 5. Hydrogel-Based Bioink Applications

Figure 6 illustrates the main applications of hydrogel-based bioinks, emphasizing soft tissue and bone tissue engineering.

### 5.1. Cartilage

Cartilaginous tissue has essential functions in the functioning of the human body, as it can support loads exerted on the joints and intervertebral discs, in addition to forming regions, such as the ears and nose [159,160,161]. It can be classified into three main types: fibrocartilage, hyaline cartilage, and elastic cartilagem [161,162]. Each is composed of different proportions of collagen and proteoglycans, which provide different biomechanical properties [162]. Furthermore, cartilaginous tissue is avascular and has difficulty self-healing [163]. Therefore, injuries affecting this tissue, such as osteoarthritis, need alternatives to ensure better treatment and quality of life [164]. Thus, 3D bioprinting has become an exciting option to circumvent this problem [161,165]. The work of Jia et al. (2022) is an example of this statement. The authors developed cartilage with high fidelity to the auricular cartilage tissues. It was possible due to the elaboration of a bioink based on gelatin methacrylate (GelMA), poly(ethylene oxide) (PEO), and polycaprolactone (PCL) interspersed with auricular chondrocytes [165,166]. Furthermore, Young et al. (2018) developed bioinks based on agarose/alginate (AG/SA) and collagen and alginate (CO/SA) mixed with chondrocytes to be used in in vitro bioprinting of cartilage tissues. The results indicated that, although the AG-/SA-based bioink showed good mechanical properties, the (CO/SA) bioinks stood out more in terms of mechanical strength and the material’s biological functionality. Therefore, bioinks from CO/SA may be promising candidates in cartilage tissue engineering [166].

### 5.2. Skin Tissue

The skin is the primary protection of the biological system against external attacks; because of this, it is injured regularly [167,168]. It is composed of three layers: the epidermis (which contains melanocytes, Merkel cells, and Langerhans cells), dermis (which is rich in fibroblasts, extracellular matrix (ECM), collagen and elastin), and hypodermis (which stores fat cells) [169]. The quest to try to biomimic the complex structure of the skin through 3D bioprinting aims not only to seek alternatives against chronic wounds [74], as in the case of burns [170], but also to develop artificial tissues that can be used for replacement of the use of animals in in vitro and in vivo tests [171]. Ng et al. (2018) developed a pigmented artificial skin through 3D bioprinting. The results showed that the skin presented a well-developed epidermis and uniform melanin distribution, which could potentially be used in toxicological tests and research focused on cell biology [172].

### 5.3. Bone Tissue

Several factors can cause defects or injuries in bones, such as aging, fractures, and infection [173]. However, the alternatives used to solve this problem, such as grafts, are limited due to poor tissue integration, autoimmune response, or donor area morbidity [83]. Thus, bioprinting presents itself as an exciting alternative to try to circumvent this problem [174], as it can provide the microstructural reconstruction of the bone structure through the use of scaffolds [175,176]. Shen et al. (2022) developed vascularized bone tissue to treat defects using in situ 3D bioprinting technology. Photocrossed extracellular matrix hydrogels, interspersed with bone mesenchymal stem cells and a thermosensitive hydrogel, were used to develop this research [176]. Im et al. (2022) formulated bioinks based on alginate, tempo-oxidized cellulose nanofibrils (TOCNF), and polydopamine nanoparticles (PDANPs) to act in bone tissue engineering. According to the authors, these bioinks could be implemented to elaborate scaffolds for tissue regeneration and the construction of artificial bone tissues [177].

## 6. Challenges and Future Prospects

Three-dimensional bioprinting is a cross-science closely related to medical sciences, biology, mechanical engineering, and materials science [19]. With unparalleled architectural control, adaptability, and repeatability characteristics, it has the potential to overcome the limits of conventional biofabrication techniques. This technology may be the most significant technological disruptor of the current design model, service delivery, health, and research. Incorporating human cells and biocompatible materials should provide a paradigm shift for surgical procedures, offering the potential to 3D print living tissues and organs and circumventing the need for organ transplantation and the use of animals in the development and testing of new drugs. Patients would also be able to access bespoke treatments [178]. However, there is still a multitude of challenges that need to be overcome. The use of the raw material, bioink, has been a primary challenge regarding structural stability, biodegradability, bioprintability, and bioactive properties, in preclinical and clinical models [179]. A limited number of bioprintable bioinks accurately represent the tissue architecture needed to restore organ function after printing. In some cases, a combination of methodological approaches is used, including heterogeneous network structures, which aim at material hydration and mechanical energy dissipation [180]. Hybrid bioinks can also be designed to amalgamate all these aspects. Furthermore, the bioprinting process itself needs to be more cell-friendly. The shear stress applied to cells during printing is detrimental to cell growth and can even alter gene expression profiles. For example, stem cells, such as iPSCs, are sensitive to such physical forces and generally do not survive the printing process. As stem cell studies have mainly been carried out in 2D environments, there are still many unknowns for a 3D stem cell culture.

## Figures and Tables

**Figure 1 jfb-13-00214-f001:**
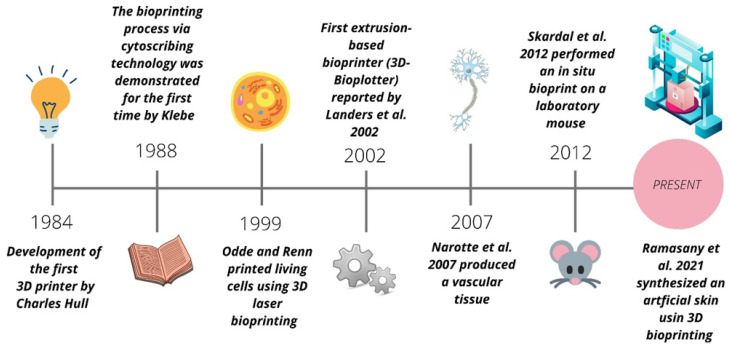
Evolution of 3D bioprinting process [20,23,24,27].

**Figure 2 jfb-13-00214-f002:**
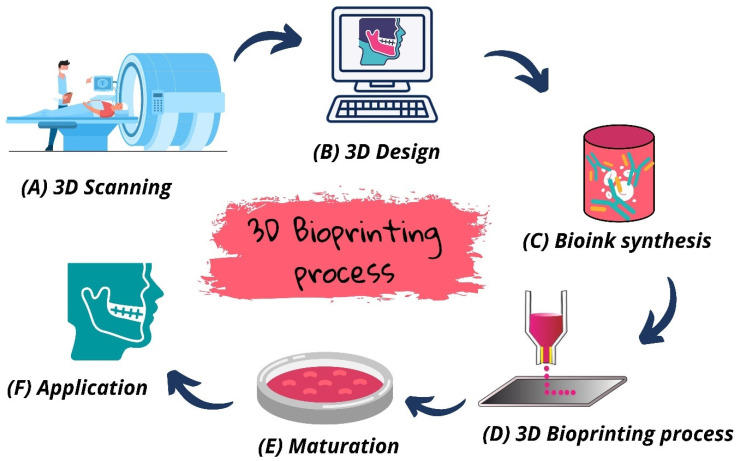
Stages of the 3D bioprinting process.

**Figure 3 jfb-13-00214-f003:**
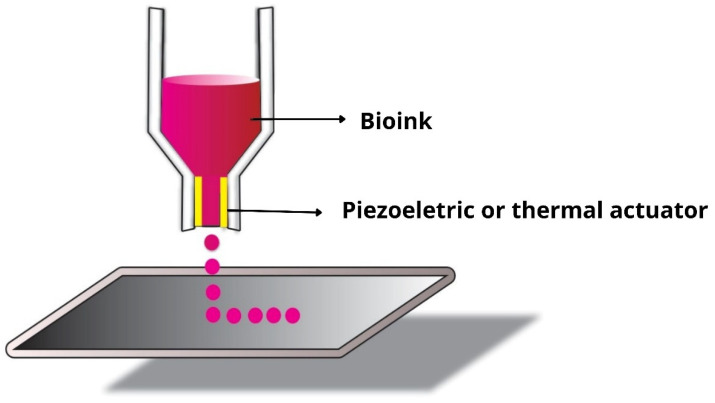
Inkjet bioprinting process.

**Figure 4 jfb-13-00214-f004:**
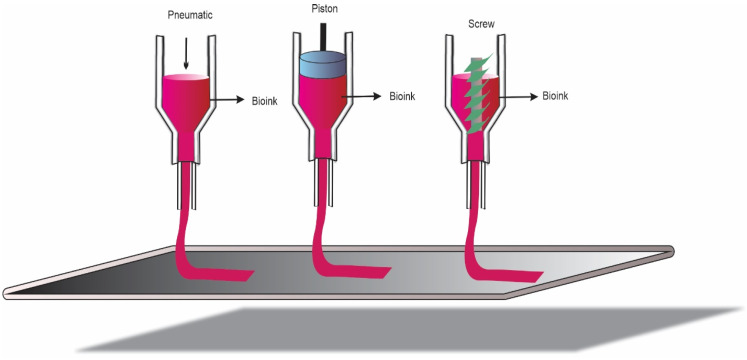
Extrusion-based bioprinting process.

**Figure 5 jfb-13-00214-f005:**
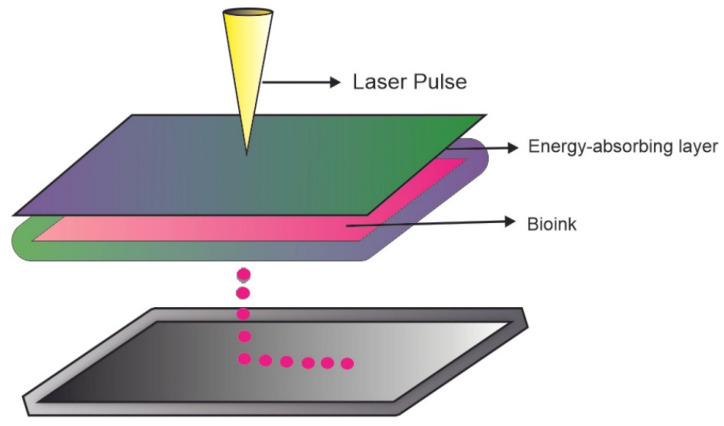
Laser assisted bioprinting process.

**Figure 6 jfb-13-00214-f006:**
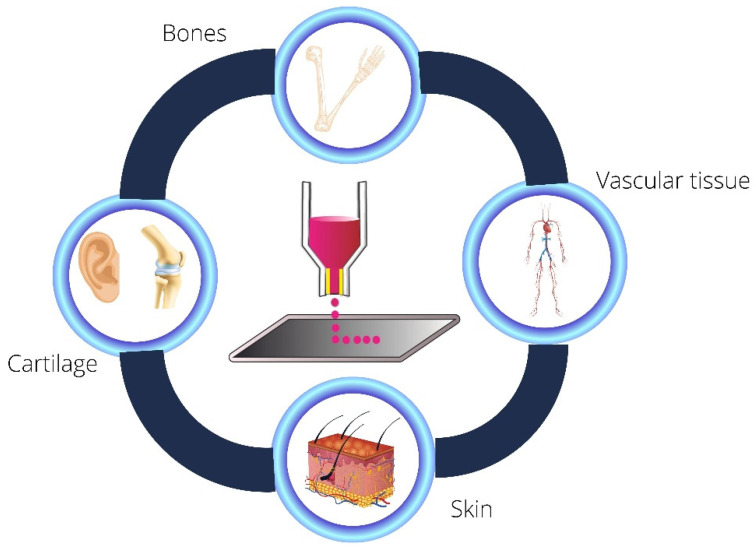
Hydrogels-based bioink applications.

**Table 1 jfb-13-00214-t001:** Comparation of three 3D Bioprinting technology.

3D Bioprinting Technology	Advantages	Disadvantages	Applications	References
Inkjet bioprinting	Low cost; High resolution and print speed; High cellular viability (>85%).	Low cell density (<106 cells mL^−1^); Bioinks with low viscosity.	Tissue regeneration; Bone; Cartilage.	[64,67,68]
Extrusion-based bioprinting	Printing bio-inks with high viscosities; High cell density; Can print various formats of materials;	Low resolution and print speed.	Cartilage; Skin; Blood vessel.	[69,70,71,72,73]
Laser-assisted bioprinting	High cell viability (>95%); Printing of bio-inks of different viscosities; High precision.	High cost; Difficulty in printing materials on a large scale.	Organ-on-a-chip; Skin; Cornea.	[6,74,75,76]

**Table 2 jfb-13-00214-t002:** Main hydrogels used in 3D bioprinting technology and their properties.

Hydrogels	Advantages	Disadvantages	Applications	References
Hyaluronic acid	Promotes cell proliferation; Maintains cartilage homeostasis; Biocompatibility.	Poor mechanical properties.	Tissue engineering; Cartilage tissue.	[16,101,102,103,104,105]
Collagen	Supports cell adhesion, differentiation and proliferation; Crosslinks with other hydrogels to increase mechanical functions.	Weak mechanical properties.	Cartilage tissue; Muscle tissue; Skin tissue.	[106,107,108]
Gelatin	Non-toxic; Biocompatibility.	Low mechanical stability.	Vascular tissue; Bone tissue; Liver tissue.	[73,109,110]
Alginate	Non-toxic; Biodegradable; Gel forming ability; Biocompatibility.	Pure alginate presents weak mechanical properties and difficulties to promote cell proliferation	Cardiovascular regeneration; Cartilage tissue; Nerve tissue.	[111,112,113,114]
Agarose	Biocompatibility; Good mechanical properties.	Limitation on cell proliferation.	Tissue engineering; Cartilage tissue engineering; Vascular tissue.	[115,116,117]
Silk-fibroin	Biodegradability; Good mechanical properties; Can be processed in different forms.	Low rheological properties.	Regenerative medicine; Tissue engineering.	[118,119]

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
