# Peer review of "3D Bioprinting Technology and Hydrogels Used in the Process"

_jfb, 2022, doi:10.3390/jfb13040214_

Round 1
Reviewer 1 Report
Dear Authors,
Thank you for the article "A short review on 3D bioprinting technology and hydrogels 2 used in the process" which was well presented with clear graphical diagrams for understanding.
However, I do have some minor suggestions which would help to improve the quality of the article.
1. There are already numerous articles published describing 3D printing and bioinks. Therefore, I feel that the authors should take it a step further by comparing the advantages and disdavantages between the different types of bioprinting technologies. It could be presented in a table format for clearer understanding.
Properties such as viscousities, printing parameters, resolution etc could be included.
2. For the hydrogels section, the authors should also consider including a comparison summary of the different biomaterials discussed so as to give the readers a clearer understanding.
For example, what is the difference between collagen and hyaluronic acid and their applications? Why do one choose collagen and not hyaluronic acid or vice versa.
3. Please remove Pluronic F-127 from the biomaterials/ bioinks as it is not used much as a bioink, instead it is more often used as a supporting framework for hydrogels due to its gelation properties.
The authors could consider shifting Pluronic F-127 under one of the bioink paragraphs.
4. The authors could also consider including images of the respective hydrogels from their quoted studies so as to give readers a better understanding.
5. "Lee et al. (2020) synthesized and sup- 224 plemented an alginate-based bioink with decellularized methacrylated extracellular ma- 225 trix (dECM) derived from bone tissues".
dECM should be discussed in its own paragraphs with the appropriate studies. Similarly, the pros and cons of dECM and the development of dECM could also be included.
Thank you.
Author Response
Dear,
The authors thank the possibility to submit a revised paper. We have edited the manuscript to the best of our knowledge in accordance with all advices of the Reviewer. Changed text in the manuscript is marked with red. The point-by-point answers to their comments are presented in the following.
Reviewer #1:
Thank you for the article "A short review on 3D bioprinting technology and hydrogels used in the process" which was well presented with clear graphical diagrams for understanding.
However, I do have some minor suggestions which would help to improve the quality of the article.
- There are already numerous articles published describing 3D printing and bioinks. Therefore, I feel that the authors should take it a step further by comparing the advantages and disdavantages between the different types of bioprinting technologies. It could be presented in a table format for clearer understanding.
Properties such as viscousities, printing parameters, resolution etc could be included.
- The authors thanked the Reviewer for this suggestion, and Table 1 was added.
- For the hydrogels section, the authors should also consider including a comparison summary of the different biomaterials discussed so as to give the readers a clearer understanding.
For example, what is the difference between collagen and hyaluronic acid and their applications? Why do one choose collagen and not hyaluronic acid or vice versa.
- The authors thanked the Reviewer for this suggestion, and Table 2 was added.
- Please remove Pluronic F-127 from the biomaterials/ bioinks as it is not used much as a bioink, instead it is more often used as a supporting framework for hydrogels due to its gelation properties.
The authors could consider shifting Pluronic F-127 under one of the bioink paragraphs.
- The authors thanked the Reviewer for this suggestion and shifted the Pluronic F-127 with silk fibroin and gelatin.
- The authors could also consider including images of the respective hydrogels from their quoted studies so as to give readers a better understanding.
- The authors are grateful for the reviewer's comments, but obtaining permission to reproduce article images was impossible due to the revision time from this article or the payment of figures.
- "Lee et al. (2020) synthesized and sup- 224 plemented an alginate-based bioink with decellularized methacrylated extracellular ma- 225 trix (dECM) derived from bone tissues".
dECM should be discussed in its own paragraphs with the appropriate studies. Similarly, the pros and cons of dECM and the development of dECM could also be included
Reviewer 2 Report
The authors describe developments in hydrogel-based inks for 3D bioprinting applications. The technologies involved in bioprinting are thoroughly explained. Section 4 discusses hydrogels as bioinks. The explanation, however, is insufficient. There should be more citations and explanations of references. Furthermore, critical discussions should be conducted in this section to improve the overall quality of the manuscript.
1. The term 'synthesis' was mentioned by the authors in figure 1. This is not the proper term for 3D printing. Use 'develop or produce' instead. Line 33, replace the term ‘reported’ with ‘developed’.
2. Latin terms such as et al, in situ, in vitro, in vivo, and so on must be italicized.
3. Since the manuscript is about hydrogels, it is not mentioned in the introduction section. Add a section outlining the latest hydrogel innovations, along with their benefits and drawbacks. Finally, summarize the author's viewpoint and prospects for hydrogel-based inks.
4. Add figures related to the subject in section 4.
5. Cite the following most recent references in the manuscript,
Extrusion based bioprinting: 10.1021/acs.macromol.2c00052
Inkjet based bioprinting: 10.1039/D2TB00442A
Laser assisted bioprinting: 10.1016/j.bprint.2022.e00194
Author Response
"Please see the attachment."

Reviewer 3 Report
As part of the research funded by the dean, the authors presented a literature review on the use of 3D bioprinting and hydrogels. The topic is very timely. Using 3D printing techniques and newly created materials, it is possible to create organs and structures that reconstruct damaged tissue. However, I have a few comments about the work, which are listed below:
The literature review within the MDPI journal looks similar to the examples presented below:
Salmi M. Additive Manufacturing Processes in Medical Applications. Materials. 2021; 14(1):191. https://doi.org/10.3390/ma14010191
Kroczek, K.; Turek, P.; Mazur, D.; Szczygielski, J.; Filip, D.; Brodowski, R.; Balawender, K.; Przeszłowski, Ł.; Lewandowski, B.; Orkisz, S.; Mazur, A.; Budzik, G.; Cebulski, J.; Oleksy, M. Characterisation of Selected Materials in Medical Applications. Polymers 2022, 14, 1526. https://doi.org/10.3390/polym14081526
Mancha Sánchez, Enrique, et al. "Hydrogels for bioprinting: a systematic review of hydrogels synthesis, bioprinting parameters, and bioprinted structures behavior." Frontiers in Bioengineering and Biotechnology 8 (2020): 776.
In my opinion, this topic cannot be treated as a short review. Anyway, I think the article's title should be corrected first. What's more, as in the reviews mentioned above, it is worth paying attention to tables, and graphical charts to show in the author's review e.g. the progress of using materials in bioprinting or what combinations of materials were used and what was printed. Of course, the authors discuss these issues in the text, but at the same time, for greater clarity of the article, it is worth compiling them in tables or other graphical charts.
Considering the review was initially well prepared by the authors. However, it needs to be refined from the text and especially the visual side (tables, charts) to accept it. The literature review is well compiled. There was not too much self-citation.
Author Response
"Please see the attachment."

Round 2
Reviewer 1 Report
I thanks the authors for making the necessary amendments.
I have no other comments/ suggestions.
Thank you
Author Response
Dear reviewer,
The authors are grateful for all recommended contributions, which undoubtedly contributed to improving the quality of the final paper.
Best regards,
Marcele Passos
Reviewer 2 Report
The authors have not taken into account the reviewers' earlier comment no. 5. Furthermore, the manuscript has not progressed significantly from the previous version. The authors did not take the reviewers' suggestions seriously! As a reviewer, I see no significant changes in the manuscript that should be considered for publication.
Please check, for example, the following recent review articles: 10.1016/j.mser.2020.100543; 10.1016/j.compositesb.2022.109895. These manuscripts' outlines are far superior, and they have cited more papers in the hydrogel sections. However, the current reviews' main field is hydrogels, which are brief and contain little information other than general information.
Author Response
Dear,
"Please see the attachment."
Best regards

Reviewer 3 Report
The authors made all the necessary corrections in accordance with my recommendations. This allowed to increase the quality and readability of the manuscript. I accept the publication.
Author Response
Dear reviewer,
The authors are grateful for all recommended contributions, which undoubtedly contributed to improving the quality of the final paper.
Best regards